# Explainable Deep-Learning-Based Gait Analysis of Hip–Knee Cyclogram for the Prediction of Adolescent Idiopathic Scoliosis Progression

**DOI:** 10.3390/s24144504

**Published:** 2024-07-12

**Authors:** Yong-Gyun Kim, Sungjoon Kim, Jae Hyeon Park, Seung Yang, Minkyu Jang, Yeo Joon Yun, Jae-sung Cho, Sungmin You, Seong-Ho Jang

**Affiliations:** 1Department of Rehabilitation Medicine, Hanyang University College of Medicine, Seoul 04763, Republic of Korea; kennyrla@hanyang.ac.kr (Y.-G.K.); sungjoonkim5674@gmail.com (S.K.); jhpark3.md@gmail.com (J.H.P.); 2Department of Rehabilitation Medicine, Hanyang University Guri Hospital, Guri 11923, Republic of Korea; yunyujun@naver.com; 3Department of Pediatrics, Hanyang University College of Medicine, Seoul 04763, Republic of Korea; jxisfriend@hanyang.ac.kr; 4Department of Computer Science, Hanyang University College of Engineering, Seoul 04763, Republic of Korea; wkdalsrb50@naver.com; 5Robotics Lab, Research and Development Division of Hyundai Motor Company, Uiwang 16082, Republic of Korea; jaesungcho82@hyundai.com; 6Fetal-Neonatal Neuroimaging and Developmental Science Center, Boston Children’s Hospital, Harvard Medical School, Boston, MA 02115, USA; 7Division of Newborn Medicine, Boston Children’s Hospital, Harvard Medical School, Boston, MA 02115, USA

**Keywords:** adolescent idiopathic scoliosis, machine learning, deep convolutional neural network, curve progression, gait analysis, hip–knee cyclogram

## Abstract

Accurate prediction of scoliotic curve progression is crucial for guiding treatment decisions in adolescent idiopathic scoliosis (AIS). Traditional methods of assessing the likelihood of AIS progression are limited by variability and rely on static measurements. This study developed and validated machine learning models for classifying progressive and non-progressive scoliotic curves based on gait analysis using wearable inertial sensors. Gait data from 38 AIS patients were collected using seven inertial measurement unit (IMU) sensors, and hip–knee (HK) cyclograms representing inter-joint coordination were generated. Various machine learning algorithms, including support vector machine (SVM), random forest (RF), and novel deep convolutional neural network (DCNN) models utilizing multi-plane HK cyclograms, were developed and evaluated using 10-fold cross-validation. The DCNN model incorporating multi-plane HK cyclograms and clinical factors achieved an accuracy of 92% in predicting curve progression, outperforming SVM (55% accuracy) and RF (52% accuracy) models using handcrafted gait features. Gradient-based class activation mapping revealed that the DCNN model focused on the swing phase of the gait cycle to make predictions. This study demonstrates the potential of deep learning techniques, and DCNNs in particular, in accurately classifying scoliotic curve progression using gait data from wearable IMU sensors.

## 1. Introduction

Adolescent idiopathic scoliosis (AIS) is a pediatric spinal disorder that affects 1–3% of adolescents aged between 10 and 18 years [1]. AIS of the spinal curve is diagnosed on the coronal plane as a Cobb angle of at least 10 degrees with any unknown causes, and two-thirds of patients experience gradual worsening during puberty [2]. The Cobb angle is the gold standard for assessing the extent of the lateral deviation of the spinal curve in evaluating the progression of AIS. Once AIS is detected, clinicians must determine whether an intervention, including bracing and correcting exercises, is warranted or if only continued observation through adolescence to adulthood is required. According to the severity and progress of AIS, treatment options include observation with regular radiography, wearable spinal bracing, or surgical correction [3,4]. However, delaying intervention to progressive scoliosis can lead to chronic back pain, severe deformity, pulmonary disorder, and psychological effects with reduced quality of life [5].

The prediction of spinal curve progression is crucial for clinicians tasked with determining appropriate treatment. Previous studies reported that spinal curve progression can be linked to multiple factors, including sex, an initial Cobb angle exceeding 25 degrees, and the growth potential of patients as indicated by lower Risser grades and menarchial status [6,7]. However, accurate prediction of curve progression based solely on the initial Cobb angle and Risser grade remains challenging due to individual variability and the complexity of progression, particularly in patients with mild scoliosis [8]. In addition, those methods only measure the static aspects of patients.

Scoliosis is associated with lateral curvature of the spine and may affect spino-pelvic mobility, potentially affecting human locomotion patterns. Studies have demonstrated that AIS often exhibits asymmetries and alterations in spatiotemporal gait parameters, such as step length and cadence, which can vary based on the severity and location of the scoliotic curve [9,10,11,12]. However, the association between gait patterns and progressive scoliosis is unclear because factors associated with scoliosis progression and gait patterns are complex and difficult to analyze. Investigation of the association between progressive scoliosis and gait patterns is therefore needed.

Emerging advancements in machine learning (ML) offer promising avenues for more precise analysis of complex gait data [13,14,15]. In particular, the SVM and RF approaches have gained prominence for their ability to accurately classify gait data from wearable sensors [15]. Deep learning (DL), a subfield of ML, excels in uncovering intricate patterns from large datasets. Of particular interest is the DCNN, a DL classifier trained on extensive image datasets and capable of discerning object features and supporting decision-making across various domains, including computer vision, speech, and language processing [16]. DL has been used to detect and predict the curve progression of AIS with high accuracy, producing correlation values using three-dimensional depth-sensor imaging or spinal radiographs [17,18,19,20,21,22,23]. These findings suggest DL techniques can enhance prognostic accuracy and inform clinical decision-making in the management of AIS. Recently, Samadi et al. investigated the use of ML to assess the severity of spinal deformity in patients with AIS by analyzing lumbosacral joint effort during gait [24]. Their study achieved an accuracy rate of 91.4%, which suggests the technique can serve as a radiation-free method of monitoring AIS progression. However, to the best of our knowledge, no research has been conducted on using an ML model to predict the progression of curvature based solely on gait data in patients with AIS. This approach may offer a radiation-free alternative that provides continuous rather than momentary assessment along with valuable insights into the biomechanical aspects of scoliosis.

Gait cyclograms provide a visual representation of closed trajectories by jointly pairing and plotting the angle of one joint against the angles of other joints over time, offering a comprehensive depiction of gait kinematics throughout the gait cycle [25]. Hip–knee (HK) cyclograms plot the hip joint angle against the knee joint angle, providing valuable information about the coordinated angular movements of multiple joints. Previously, Park et al. and Lee at al. analyzed gait features using HK cyclograms of patients with knee osteoarthritis and stroke, respectively [26,27]. Gait analysis using cyclograms and ML could offer a more comprehensive approach to assessing gait abnormalities in AIS patients and scoliosis progression.

Here, we introduce a novel approach utilizing gait features and HK cyclograms derived from inertial measurement unit (IMU)-based analysis of gait in patients with AIS as input for ML algorithms. Our primary objective was to develop an accurate ML model for classifying progressive and non-progressive scoliotic curves based on gait features in AIS patients. Additionally, we aimed to evaluate and compare the predictive performance of various ML models regarding scoliotic curve progression. By leveraging these techniques on novel gait data representations, we sought to provide a robust method of assessing curve progression likelihood through gait analysis, facilitating informed treatment decisions and personalized patient management strategies.

## 2. Materials and Methods

### 2.1. Design

This study was approved by the Institutional Review Board (IRB) of the Hanyang University Guri Hospital (IRB File No. 2022-08-025). We retrospectively reviewed data from AIS patients who visited or were referred to our hospital between January 2017 to December 2021. This study involved 38 AIS patients who underwent spine radiography and gait trials. Patients were classified into progression and non-progression groups based on Cobb angle changes per month. Gait analyses were conducted using an IMU-sensor-based system, and ML models—including SVM, RF, and DCNN—were employed for prognosis prediction. Statistical analyses and model evaluations were performed to compare patient characteristics and assess model performance.

### 2.2. Subjects

All patients received spine radiography at the first and second visit and completed a gait trial at the first visit. The inclusion criteria were (1) suspected AIS at the first visit, (2) a Cobb angle above 10 degrees on full-spine X-rays at the first or second visit, and (3) no applied brace treatment nor spine surgery during the next visit. Only the primary curve (the greatest Cobb angle) was examined in cases when patients had multiple curvatures such as thoracolumbar scoliosis. Patients with missing clinical data or those who were lost to follow-up were excluded.

Patients were classified into two groups according to the change in Cobb angle per month: a progression group (P) and a non-progression group (NP). Due to the variation of visiting terms for each patient, we defined progression as an increase in the Cobb angle of more than 0.3 degrees per month [7]. The covariates assessed in this study included age, weight, height, height change per month, follow-up period, the first Cobb angle as continuous variables, and the Risser grade as a non-continuous variable.

### 2.3. Instrumentation

The inertial measurement unit (IMU)-sensor-based gait analysis system (Human Track, R. Biotech Co., Ltd., Seoul, Korea) used in this study was validated in previous studies [26,27,28,29]. Seven IMU sensors were positioned on the lower abdomen, the middle of both femurs, the shafts of both tibias, and the dorsum of both feet, as shown in Figure 1. All participants were instructed to navigate a 10 m gait course at a self-determined walking speed after becoming accustomed to walking with the IMU sensors. Tri-axial acceleration, angular velocity, and magnetometer readings were obtained from the seven IMU sensors during the walking trials at a frequency of 100 Hz. A prior method’s offset value and gain compensation value were used to reduce errors in the accelerometers and gyroscope [30]. Data from trials involving hindrances on the pathway, equipment misalignments, or abrupt halts were excluded.

### 2.4. Hip–Knee Cyclogram

All HK cyclograms were generated by concurrently plotting the angles of the hip and knee joints throughout the entire gait cycle on the sagittal, transverse, and coronal planes [25,31]. The HK cyclograms were plotted in a clockwise direction, progressing from the stance phase to the swing phase, as determined by the respective heel strike and toe-off points and the angular velocity of the foot dorsum [26,27]. We then calculated the average values of the HK cyclogram parameters for each plane, including range of motion for hip and knee joint angles and geometric features such as center of mass, the means for hip and knee joint angles and perimeter (stance, swing, and total), and area (stance, swing, and total).

As depicted in Figure 2, cyclograms comprise successive data points. The perimeter was the linear summation of the lengths of the lines connecting the data points while the area represented the space enclosed within this perimeter. The perimeter was calculated using the following equations:
(1)$$L_{i}=\sqrt{{{(\theta }_{h_{i}}-\theta_{h_{i+1}})}^{2}+{{(\theta }_{k_{i}}-\theta_{k_{i+1}})}^{2}}=∆t\sqrt{{\omega_{h_{i}}}^{2}+{\omega_{k_{i}}}^{2}}$$
(2)$$L=\sum_{i}L_{i}$$
(3)$$\mathrm{Area}=\frac{1}{2}\sum_{i}{(\theta }_{h_{i}}\theta_{k_{i+1}}-\theta_{h_{i}+1}\theta_{k_{i}})$$

$\theta_{h_{i}}$ and $\theta_{k_{i}}$ represent the hip and knee joint angles, respectively, of point *i*. $\omega_{h_{i}}$ and $\omega_{k_{i}}$ represent the average angular velocities of the hip and knee joints, respectively, at a specific time interval, Δ*t*. The perimeter was calculated as L, which is the summation of $L_{i}$ and this parameter provides insight into the average joint velocity. The area enclosed by the set of successive data points was calculated using Equation (3), using the combined range of joint movements [32,33].

### 2.5. Machine Learning Model

In this study, we applied two approaches to predicting a prognosis of scoliosis. First, we applied conventional (i.e., SVM and RF) ML algorithms as reference models based on 36 handcrafted features from the HK cyclogram and 4 clinical factors, including age, weight, height, and initial Cobb angle [34,35]. We also proposed a DCNN that could handle the HK cyclogram as a sequential input with and without the clinical factors. To train and evaluate the model and to gain reliable estimates of the model’s generalization error, we performed k-fold cross-validation, which divides the training dataset into k parts without re-entry and uses k − 1 parts for model training and 1 part for testing. To obtain the k models and performance estimates, this technique was repeated k times. We opted for k = 10, as it has been empirically observed to provide test error rate estimates that avoid both excessively high bias and very high variance. Cross-validation was conducted subject-wise to avoid overestimating performance, while the classification models were trained stepwise. Majority voting was used to determine the final predictions for each subject after stepwise classification for each gait cycle within a gait record [36].

#### 2.5.1. Support Vector Machine

An SVM is a type of supervised learning model that uses specific learning algorithms to identify a support vector [34]. A support vector is an ideal hyperplane that effectively divides data points into binarized groups classification. An SVM determines the hyperplane with the maximum margin by identifying two data points from two different groups such that the hyperplane has the shortest distance. An SVM can be implemented in a linear or nonlinear manner. A nonlinear SVM produces superior results when the linear-margin hyperplane yields a poor match. Using an SVM classifier, we computed the distance between data from P and NP group subjects, creating a model capable of classifying each cluster based on distance.

#### 2.5.2. Random Forest

An RF is a tree-based classification method that uses numerous decision trees and is versatile in its application. An RF generates a multitude of decision trees and predicts a new object based on specific attributes. Each tree in the forest provides its categorization result and vote, with the total output of the forest being the most prevalent classification. In regression scenarios, the RF output represents the average of all decision tree outputs.

#### 2.5.3. Deep Convolutional Neural Network

We proposed a deep convolutional network model to predict the prognosis of scoliosis based on the entire pattern of the HK cyclogram along with the gait cycle. The proposed network consists of 16 residual blocks with two convolutional layers per block, including rectified linear unit activation layers, which are similar to residual network architecture [37]. The global average pooling layer is connected to the last residual block. Batch normalization and dropout layers are inserted between convolutional layers to regularize training and prevent overfitting [38,39]. Multi-plane HK cyclograms are fed to the proposed model after Z-score normalization in the input shape (batch, time, channel), where 256 and 6 are specified for the time and channel, respectively. For a model with clinical factors, four scalar values are supplied as auxiliary inputs and concatenated to the output of global average pooling layer before the last fully connected layers. A fully connected layer with softmax activation is used as the last layer to produce logits for binary classification output. In this study, the hyper-parameters of each network architecture and optimization algorithm were empirically selected through a grid search and manual fine-tuning. A 10-fold training process was performed for 100 epochs using Adam [40]. The batch size for training was 8 and the initial learning rate was 0.001, and the learning rate was updated during training by multiplying the previous learning rate by 0.9 with 5 epochs of patience. All algorithms were implemented using Python 3.8 with the TensorFlow library [41].

#### 2.5.4. Gradient-Based Class Activation Mapping

After training the deep convolutional network model, we applied gradient-based class activation mapping (grad-CAM) to the proposed network to determine how the network predicted prognoses from the HK cyclogram [42]. Grad-CAM explains the output of a deep neural network by visualizing a class activation map based on gradient-based localization. The class activation map uses the gradients of prediction to produce a coarse localization map that highlights important regions in the input for classifying the target class. In general, a class activation map is used for two-dimensional images, but we modified and applied grad-CAM to our models, which handle the HK cyclograms as input to determine which gait characteristics are meaningful in predicting a prognosis. Successfully and unsuccessfully classified groups were then compared regarding identification of the areas with the most pronounced signals in the heat map of the hip- and knee-joint gait cycle of each plane.

### 2.6. Statistical Analysis and Model Evaluation

Statistical analyses were conducted to compare patient characteristics between the progression and non-progression groups. Continuous variables were analyzed using an independent *t*-test, while non-continuous variables were compared using Fisher’s exact test. A *p*-value < 0.05 was considered statistically significant. All analyses were performed using SPSS software version 26.0.

The prediction performance of binary classifiers was evaluated by calculating accuracy, precision, recall, and F1 score as follows:Accuracy = TP + TN/FP + FN + TP + TN(4)
Precision = TP/TP + FP(5)
Recall = TP/TP + FN(6)
F1 score = 2 × (Recall × Precision)/Recall + Precision(7)
where TP is the number of true positives that the classifier predicted, TN is the number of true negatives, FP is the number of false positives, and FN is the number of false negatives.

We also computed the area under the receiver operating characteristics (ROC) curve to compare the classification performance of each ML model. The area under the curve (AUC) exhibits greater convergence than accuracy and shows the average sensitivity over all conceivable specifications. The AUC is calculated using the trapezoid approach based on the ROC curve, which graphically depicts the tradeoff between the TP and FP rates.

## 3. Results

The demographic and clinical characteristics of the study population are presented in Table 1. A total of 14 and 24 patients were classified into the P and NP groups, respectively. The initial Cobb angles of the P and NP groups were not significantly different. There were no significant differences between the two groups in height, weight, growth rate, or Risser grade. However, there was a significant difference between the groups in the second Cobb angle and the change in Cobb angle per month.

The prediction performance of each model can be shown by calculating the accuracy, precision, recall, and F1 score for subject-wise binary classification (Table 2). A DCNN model using multi-plane HK cyclograms and clinical factors achieved an accuracy of 92%, a precision of 91%, a recall of 92%, and an F1 score of 92%, indicating superior performance compared to DCNN, which used only multi-plane HK cyclograms (84% accuracy, 85% precision, 84% recall, and 84% F1 score). This was superior to the SVM using geometric features and clinical factors (55% accuracy, 61% precision, 55% recall, and 58% F1 score) and the RF using geometric features and clinical factors (52% accuracy, 42% precision, 43% recall, and 42% F1 score). These results show the usefulness of multi-plane HK cyclograms as sequence inputs for DL rather than using handcrafted features computed from them. 

We also compared the performance of each model in a sample-wise manner before applying majority voting (Table 3). Similar to the subjectwise classification results, in stepwise classification, the DCNN model with multi-plane HK cyclograms with clinical factors achieved the best performance, with an accuracy of 83%, a precision of 81%, a recall of 82%, and an F1 score of 82%. The ROC curve with the AUC of each binary classification model is shown in Figure 3. The AUC of the SVM, RF, and DCNN without and with clinical factors are 0.57, 0.73, 0.76, and 0.80, respectively.

The grad-CAM heatmap includes regions that the DCNN model prioritized when predicting scoliosis progression with HK cyclograms of each multi-plane (Figure 4 and Figure 5). Grad-CAM predicted scoliosis progression by focusing on the swing phase of the cyclogram.

## 4. Discussion

In our study, binary classification models employing ML algorithms achieved reasonable degrees of accuracy when tasked with distinguishing between progressed and non-progressed curvature based on gait features derived from IMU-derived data in patients with AIS. The most effective model was a DCNN using clinical factors and multi-plane HK cyclograms (accuracy = 92%, F1 score = 92%, AUC = 0.80), which represent two-dimensional inter-joint coordination of gait data as input features. Other ML classifiers, SVM and RF with geometric features, achieved less satisfactory results (accuracy = 55%, AUC= 0.57; accuracy = 68%, AUC = 0.73, respectively). These findings show the potential of ML techniques, particularly DCNN, in accurately predicting the progression of scoliotic curvature based on gait analysis. Traditional gait analysis encounters challenges due to the extensive dataset and numerous interdependent parameters, making interpretation difficult [43]. In this respect, conventional ML techniques, such as an SVM and a RF, can effectively handle and analyze the large and complex datasets from gait analysis [44,45]. DL models, and deep neural networks or DCNNs in particular, can automatically learn and extract relevant high-level features from raw gait data, eliminating the need for manual feature engineering [46,47]. This is advantageous over conventional ML methods, which typically require handcrafted features as input. Additionally, while conventional ML methods offer some interpretability through feature-importance scores, DL can provide more comprehensive and visually interpretable explanations for a model’s predictions and decision-making process [48,49]. This allows for an understanding of the model’s decision-making process, addressing the black-box nature of conventional ML methods. A DL method may therefore offer valuable insights for clinical decision-making and patient management strategies. No previous study has applied DL to an analysis of gait data for AIS, including the HK cyclogram that is high-dimensional visualized data. Our study confirmed the accuracy of the DCNN model in classifying gaits between cases of progressed and non-progressed AIS using data derived from IMU sensors.

The identification of risk factors associated with the progression of scoliotic curves is an important topic of research, but results to date have been inconsistent. Lonstein et al. proposed a nomogram with a formula for calculating the risk of progression, incorporating factors such as the magnitude of the curve, Risser grade, and chronological age, which were deemed most relevant for progression [7]. Charles et al. identified higher curve-progression velocity and a greater Cobb angle at the onset of puberty as pertinent risk factors for curve progression [50]. Tan et al. reported that an initial Cobb angle exceeding 25 degrees achieved the best AUC value of 0.80 for progression exceeding 30 degrees at skeletal maturity [51]. While these indicators were significantly correlated at higher magnitudes, their predictive utility for mild and moderate stages remains uncertain. Additionally, repetitive measurement of the Cobb angle and assessments of skeletal maturity entail frequent exposure to radiation, which poses health risks to patients [52]. The conventional method of evaluating the risk of progression for AIS is limited by high variability among clinicians, and measurement of static features is susceptible to variations in posture [53]. Integrated approaches that incorporate dynamic factors, such as daily habits of walking and posture, alongside traditional radiological assessments may offer a more holistic picture of the risk landscape associated with AIS progression.

Our proposed approach offers a promising avenue to mitigate exposure to radiation during puberty while dynamically analyzing the biomechanical characteristics of patients with AIS using gait data from IMU sensors. Previous studies have explored ML algorithms with spine radiographs or three-dimensional parameters to classify progressive spinal curves in AIS. For example, Yahara et al. and Wang et al. used DCNNs to predict progression in AIS, achieving accuracies of 69% and 76.6%, respectively, by analyzing plain spine radiographs [18,23]. Nault et al. and Garcia-Cano et al. employed general linear models and RF classifiers, respectively, to analyze three-dimensional parameters of scoliotic curves obtained from spine radiographs to predict curve progression [54,55]. However, these approaches rely primarily on conventional static indicators as inputs. As a result, they may not offer optimal rehabilitation treatment for individuals with AIS. By contrast, our approach leverages dynamic gait data captured through IMU sensors, providing a more comprehensive understanding of the biomechanical characteristics of AIS patients.

To the best of our knowledge, our study is the first to analyze the relationship between curve progression in AIS and gait features using an explainable DL model. While several studies have explored the impact of scoliosis on gait patterns from a biomechanical perspective, our approach uses ML algorithms to classify the progression of spinal curvature in AIS patients based on gait features. For example, Choi et al., who used an artificial neural network to analyze gait asymmetry in AIS, achieved a relatively high accuracy of 96.3% with a selected set of 11 input variables out of 60 gait variables [56]. Similarly, Samadi et al., who used ML algorithms based on lumbosacral joint efforts acquired from a gait kinematic analysis to classify the severity of spinal deformity in AIS, reported promising results with an ensemble classifier that achieved an accuracy of 91.4% [24]. However, previous studies often relied on camera-based optokinetic gait analysis, which is relatively expensive and spatially inefficient compared with our wearable IMU sensors [57]. Cho et al. used an SVM to classify control and AIS groups based on IMU-sensor-based gait data, achieving a high accuracy of 90.5% [58]. Although their method was 85.7% accurate in classifying the severity of scoliosis, they did not specifically focus on the risk of curve progression in AIS patients but on comparing the gait characteristics of AIS and normal individuals or within different severity levels of AIS patients. Our study aims to fill this gap by specifically addressing the relationship between gait features and curve progression in AIS patients, providing valuable insights into the potential of ML techniques to predict and manage AIS progression.

A grad-CAM heatmap revealed the specific regions prioritized by the DCNN when predicting scoliosis progression using multi-plane HK cyclograms. The analysis indicated that the DCNN focused predominantly on the swing phase of the cyclogram to make predictions about scoliosis progression. In Figure 5, HK cyclograms are partitioned into hip- and knee-joint angles to reveal the intervals highlighted by grad-CAM across the gait cycle. The highlighted intervals occurred mainly around the swing phase, particularly during the pre-swing, initial swing, and mid-swing phases. This suggests that the kinematic features extracted during the swing phase can help identify patients at risk of scoliotic curve progression. In previous studies, during the swing phase of the gait cycle, patients with AIS exhibited distinct gait characteristics. Liu et al. reported that AIS patients have a longer swing phase (38.66 ± 0.97 s vs. 37.62 ± 1.08 s) and smaller maximum pressure peak in the gait cycle compared with healthy subjects [59]. Furthermore, trunk-tilting angles at the end of the swing phase were larger in AIS patients compared to age-matched controls [60]. Cho et al. reported that hip joint movement in patients with scoliosis is important due to significant differences in the flexion and extension of the convex side hip during the swing phase compare with normal conditions [58]. These biomechanical changes in the swing phase of AIS patients may be correlated with the results of grad-CAM. However, no study has reported a significant difference, particularly during the pre-swing, initial swing, and mid-swing phases of gait in progressed AIS, indicating a need for further investigation. Grad-CAM calculates gradients by back-propagating prediction scores to the target layer of the network and integrating forward feature maps by fitting them as weights. However, these weights may inadequately assess the significance of the feature map, which itself may contain noise and incorporate non-essential components [61].

There are several limitations to our findings. First, we analyzed the gait data of 38 patients with AIS who visited a single center. Utilizing data from a limited sample size at a single center could lead to biases and limit the generalizability of the findings. Larger and more diverse datasets from multiple centers would be beneficial for developing more accurate and widely applicable models while also mitigating the risk of overfitting and bias. Employing techniques such as k-fold cross-validation and dropout regularization helps reduce overfitting, but broader datasets would enhance model robustness. Second, the definition of progression could introduce selection bias, especially considering variations in the duration between visits for different subjects. These variations can distort the assessment of curve angle changes, potentially leading to biased classifications. Future studies should refine the criteria for progression to account for differing follow-up durations and mitigate such biases. Furthermore, while the study evaluated SVM, RF, and DCNN classifiers, other algorithms such as K-nearest neighbor or DL structures could also be explored. A broader range of algorithms and a larger dataset could provide insights into their comparative performance and identify the most suitable models for AIS prognosis predictions. While grad-CAM analysis provided some insights into the regions of the HK cyclogram that the DCNN model focused on, interpreting DL models remains challenging. Further research is needed to better understand the decision-making process of these complex models and the specific gait features driving their predictions. Finally, conducting gait analysis without regulating walking speed, such as on a treadmill, may have introduced variability to the gait parameters. While gait speed influences gait kinematics significantly, natural, everyday walking better reflects real-life scenarios. The minimal influence of gait speed on HK cyclograms observed in this study suggests that the impact may be limited. Addressing these limitations in future research will help refine and advance our understanding and application of IMU-sensor-based gait analysis and ML methods in assessing ambulatory function and predicting disease progression in musculoskeletal disorders such as AIS.

## 5. Conclusions

The findings of this study demonstrate the potential of ML approaches, particularly DCNNs using gait analysis data from wearable inertial sensors to predict scoliotic curve progression in patients with AIS. The DCNN model, which used multi-plane HK cyclograms and clinical factors, outperformed conventional ML models such as the SVM and RF, achieving an accuracy of 92%, precision of 91%, recall of 92%, and F1 score of 92%. The model focused on the swing phase of the gait cycle, aligning with previous studies reporting gait alterations in AIS. Further research with larger datasets and refined criteria is needed to enhance generalizability. Overall, this study highlights the value of integrating clinical factors, gait analysis from wearable sensors, and ML for predicting and managing musculoskeletal disorders like AIS, paving the way for personalized, data-driven patient care.

## Figures and Tables

**Figure 1 sensors-24-04504-f001:**
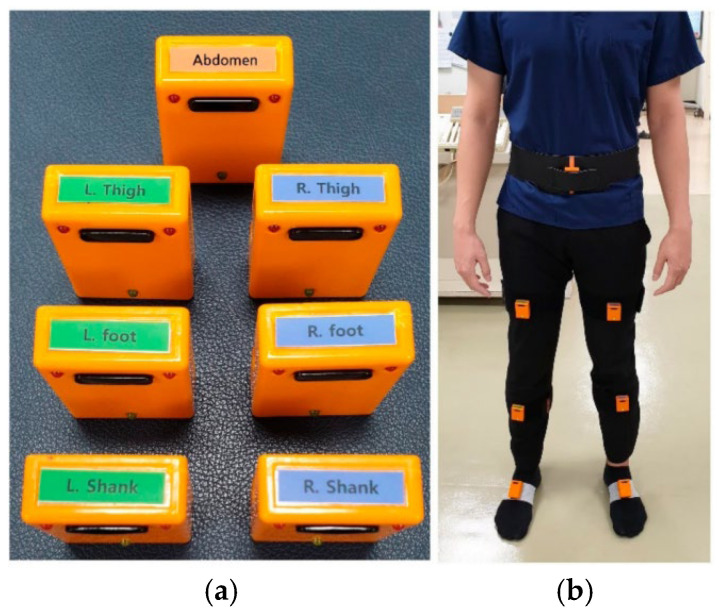
(**a**) IMU sensors used in the gait analysis system and (**b**) IMU sensors on the dorsum of both feet, the shafts of both tibias, the middle of both femurs, and the lower abdomen. This “Figure 1” by Lee et al. is licensed under CC BY 4.0 [27].

**Figure 2 sensors-24-04504-f002:**
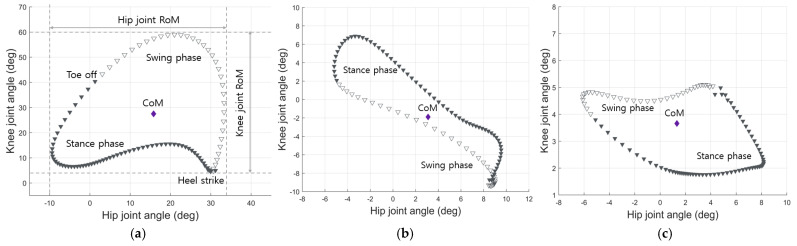
Representative multi−plane hip−knee cyclograms. Hip and knee joint angles (degrees) during the gait cycle are plotted in the clockwise direction on the x and y axes, respectively. The gait cycle is divided into stance (filled inverted triangles) and swing (open inverted triangle) phases: (**a**) sagittal plane; (**b**) transverse plane; (**c**) coronal plane. RoM = range of motion; CoM = center of mass.

**Figure 3 sensors-24-04504-f003:**
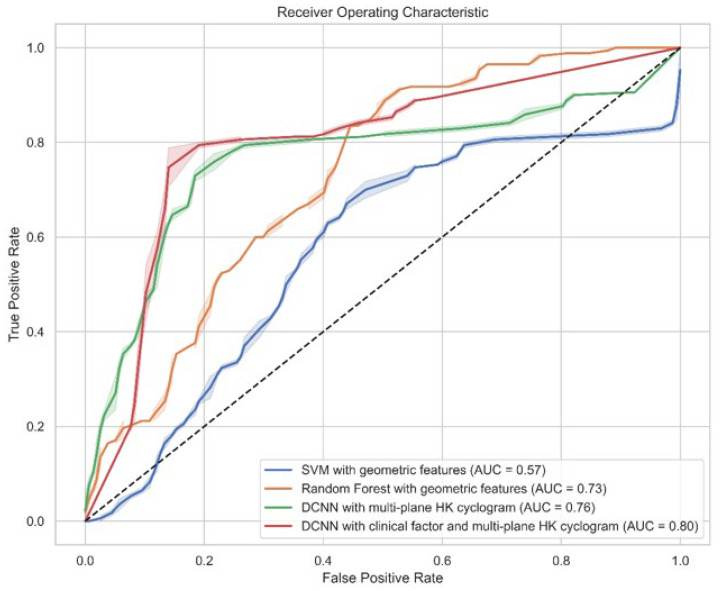
Receiver operating characteristic curve with the AUC for each method. A DCNN with clinical factor and multi-plane HK cyclogram (AUC = 0.80) showed the best value. AUC = area under curve; HK cyclogram = hip–knee cyclogram; SVM = support vector machine; DCNN = deep convolutional neural network.

**Figure 4 sensors-24-04504-f004:**
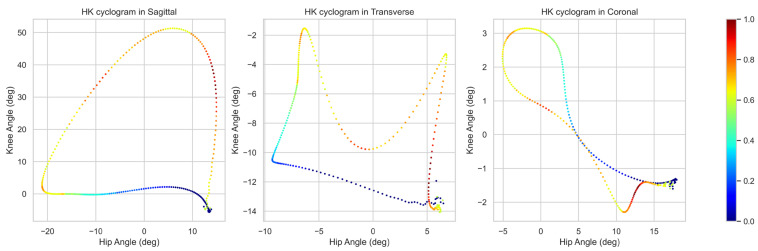
Grad-CAM visualization of the prognosis prediction model over HK cyclograms. Intervals highlighted in red are relevant to the progression group to which the DCNN paid the most attention when classifying between groups. Grad-CAM = gradient-based class activation mapping; HK cyclogram = hip–knee cyclogram; DCNN = deep convolutional neural network.

**Figure 5 sensors-24-04504-f005:**
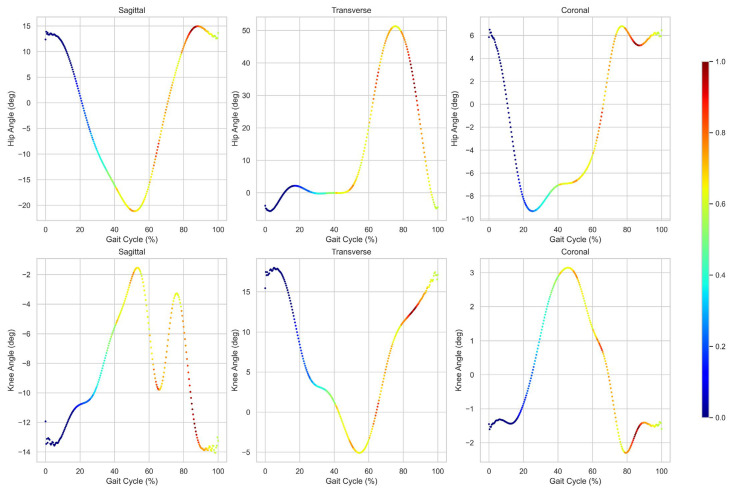
Grad-CAM visualization of the DCNN model with separated hip and knee angles, respectively, in the sagittal, transverse, and coronal planes. Grad-CAM = gradient-based class activation mapping; DCNN = deep convolutional neural network.

**Table 1 sensors-24-04504-t001:** Demographic and clinical data. The results are presented as mean ± standard deviation, or n (%). *p*-values are calculated by independent *t*-test and Fisher’s exact tests *.

	Progression (P) (*n* = 14)	Non-Progression (NP) (*n* = 24)	*p*-Value
Age (years)	12.9 ± 2.5	11.9 ± 2.5	0.217
Sex (M:F)	6:8	9:15	
Height (cm)	155.1 ± 8.8	147.9 ± 11.8	0.055
Weight (kg)	49.9 ± 10.5	44.2 ± 11.0	0.138
Height change per month	0.30 ± 0.25	0.56 ± 0.45	0.060
Risser grade *			0.127
grade 1–3	8 (57.1%)	20 (83.3%)	
grade 4–5	6 (42.9%)	4 (16.7%)	
Scoliosis type			
Thoracic	4	11	
Lumbar	2	5	
Thoracolumbar	8	8	
Follow-up period (months)	11.9 ± 6.7	11.7 ± 9.3	0.967
First Cobb angle	13.0 ± 6.9	11.6 ± 4.4	0.476
Second Cobb angle	16.9 ± 6.9	8.6 ± 5.5	<0.001
Cobb angle change per month	0.35 ± 0.19	−0.40 ± 0.38	<0.001

Abbreviations: *n* = number of participants; M = male; F = female.

**Table 2 sensors-24-04504-t002:** Subject-wise comparison of the scoliosis prognosis prediction performance.

Methods	Accuracy	Precision	Recall	F1 Score
SVM with geometric features	55%	50%	50%	49%
RF with geometric features	68%	65%	67%	65%
DCNN with multi-plane HK cyclograms	84%	82%	84%	83%
DCNN with clinical factors and multi-plane HK cyclograms	**92%**	**91%**	**92%**	**92%**

Abbreviations: SVM = support vector machine; RF = random forest; DCNN = deep convolutional neural network; HK cyclograms = hip–knee cyclograms.

**Table 3 sensors-24-04504-t003:** Step (sample)-wise comparison of the scoliosis prognosis prediction performance.

Methods	Accuracy	Precision	Recall	F1 Score
SVM with geometric features	59%	53%	53%	53%
RF with geometric features	67%	63%	62%	62%
DCNN with multi-plane HK cyclograms	78%	76%	77%	77%
DCNN with clinical factors and multi-plane HK cyclograms	**83%**	**81%**	**82%**	**82%**

Abbreviations: SVM = support vector machine; RF = random forest; DCNN = deep convolutional neural network; HK cyclograms = hip–knee cyclograms.

## Data Availability

The data presented in this study are available upon request from the corresponding authors.

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
