# Peer review of "Explainable Deep-Learning-Based Gait Analysis of Hip–Knee Cyclogram for the Prediction of Adolescent Idiopathic Scoliosis Progression"

_sensors, 2024, doi:10.3390/s24144504_

Round 1

Reviewer 1 Report

Comments and Suggestions for Authors

In this study, the authors propose machine learning models for the classification of progressive and non-progressive scoliotic curves based on gait analysis with wearable inertial sensors in adolescent idiopathic scoliosis.  In particular, they show the potential of deep learning techniques, and DCNN in particular, in the accurate classification of scoliotic curve progression.

A few considerations:

- Some abbreviations have already been introduced in the abstract there is no need to reintroduce them (e.g. lines 73 and 76).

- The initials of the different classifiers should be uppercase (lines 30-31, 72-73 and 76).

- It would be a good idea to introduce the acronym Deep Learning without repeating it several times.

- The obtained performance of the metrics must all be reported either as percentages or as decimals (for example lines 265-267)

Comments on the Quality of English Language

Line 25-26 unify verbs (past and present tense)

Reviewer 2 Report

Comments and Suggestions for Authors

Thank you very much for inviting me to review this interesting manuscript on predicting AIS progression using AI models. While the study is interesting, I have some concerns about the methodology. In particular, the study’s sample size is small, which reduces the external validity of the work. I have further explained my concerns below.

Major comments

a.      Only 38 participants were included in the study, which makes me concerned about the external validity of the work.

b.      I suggest adding a design subheading at the beginning of the methods section and describing the relevant information (e.g., location, date, and study design) there.

c.      As all participants were younger than 18, how did you ensure the ethical aspects of the work? In particular, parental consent was needed to include the patients in the study.

d.      #1 eligibility criterion is a bit vague to me. In fact, participants with definite diagnoses should have been included in the study rather than those merely suspected of having AIS. I understand that you have further evaluated the patients and confirmed the diagnosis; however, this particular criterion is a bit misleading.

e.      In the methods section, please briefly describe how the covariates were assessed in your study.

f.       The conclusion section repeats the results. I suggest revising it and focusing on the take-home messages of the study.

Minor comments

a.      Please define the abbreviations in the tables’ footnotes. 

Round 2

Reviewer 2 Report

Comments and Suggestions for Authors

Thank you for revising the manuscript according to the comments. I understand the limitations related to obtaining a large sample size for such analyses. However, despite acknowledging these limitations, I believe they still impact the findings and reduce the validity of the work.